subjective well-being; WHO-5; Arab countries; Arabic; psychometric properties; cross-country validity

**Corresponding author:**
Souheil Hallit;
Email: souheilhallit@usek.edu.lb

# Cross-country validation of the Arabic version of the WHO-5 Well-Being Index in non-clinical young adults from six Arab countries

Feten Fekih-Romdhane[1,2], Wissal Cherif[1,2], Amthal Alhuwailah[3], Mirna Fawaz[4], Hanaa Ahmed Mohamed Shuwiekh[5], Mai Helmy[6], Ibrahim Hassan Mohammed Hassan[7], Abdallah Y Naser[8], Btissame Zarrouq[9], Marianne Chebli[10], Yara El Frenn[10], Gabriella Yazbeck[10], Gaelle Salameh[10], Ayman Hamdan-Mansour[11], Eqbal Radwan[12], Abir Hakiri[1], Sahar Obeid[13], Majda Cheour[1,2] and Souheil Hallit[10,14,15]

[1]Faculty of Medicine of Tunis, Tunis El Manar University, Tunis, Tunisia; [2]Department of Psychiatry Ibn Omrane, Razi Hospital, Manouba, Tunisia; [3]Department of Psychology, Kuwait University, Kuwait, Kuwait; [4]Nursing Department, Faculty of Health Sciences, Beirut Arab University, Beirut, Lebanon; [5]Department of Psychology, Fayoum University, Faiyum, Egypt; [6]Psychology Department, College of Education, Sultan Qaboos University, Muscat, Oman; [7]South Valley University, Qena, Egypt; [8]Department of Applied Pharmaceutical Sciences and Clinical Pharmacy, Faculty of Pharmacy, Isra University, Amman, Jordan; [9]Faculty of Medicine and Pharmacy, Laboratory of Epidemiology and Research in Health Sciences, Université Sidi Mohammed Ben Abdellah, Fez, Morocco; [10]School of Medicine and Medical Sciences, Holy Spirit University of Kaslik, Jounieh, Lebanon; [11]School of Nursing, University of Jordan, Amman, Jordan; [12]Department of Biology, Faculty of Science, Islamic University of Gaza, Gaza Strip, Palestine; [13]Social and Education Sciences Department, School of Arts and Sciences, Lebanese American University, Jbeil, Lebanon; [14]Psychology Department, College of Humanities, Effat University, Jeddah, Saudi Arabia and [15]Applied Science Research Center, Applied Science Private University, Amman, Jordan

## Abstract

This study aimed to perform a cross-country validation of the Arabic version of the World Health Organization 5-item (WHO-5) Well-Being Index, in terms of factor structure, composite reliability, cross-gender measurement invariance and concurrent validity. We carried out a cross-sectional, web-based study on a total of 3,247 young adults (aged 18–35 years) from six Arab countries (Tunisia, Lebanon, Egypt, Jordan, Morocco and Kuwait). Confirmatory Factor Analysis showed that the one-factor model demonstrated acceptable fit across all six countries. In addition, the Arabic WHO-5 Well-Being Index yielded high reliability coefficients in samples from each country (McDonald's $\omega$ and Cronbach's $\alpha$ = .92–.96), across genders ($\omega$ = .95 in men and .94 in women) and age groups ($\omega$ = .94/$\alpha$ = .94 in participants aged ≤25 years and $\omega$ =.96/$\alpha$ =.96 in those aged ≥26 years). Multi-group analyses demonstrated that configural, metric and scalar invariance were supported across gender, countries and age groups. Regarding concurrent validity, WHO-5 Well-being scores were strongly and significantly inversely correlated with depression, anxiety, stress, suicidal ideation and insomnia severity. This study provides a brief, valid and reliable Arabic version of the WHO-5 Well-Being Index that can be applied cross-nationally among Arabic-speaking young adult populations for screening and research purposes.





## Impact statement

There is growing recognition of significant psychological distress among Arab populations, underscoring the need for contextualized and culturally sensitive prevention approaches that focus on subjective well-being (SWB) in Arab countries. However, the cross-country validity of well-being scales remains to date underexplored in the region. The current study is the first to explore the cross-country validity of the WHO-5 Well-Being Index among young adults in six Arab countries across the Middle East and North Africa region. Our findings demonstrated that the scale's unidimensional structure was consistent across genders, age groups and respondents from different nations. Convergent and divergent validity were good, and reliability was excellent. Overall, these findings suggest that the Arabic WHO-5 measures the originally intended SWB construct in the specific context conditions of Arabic-speaking populations. By establishing the cross-country validity of the Arabic WHO-5, this study supports its broader application in epidemiological research to explore SWB among Arabic-speaking young adults across diverse geographic areas.

## Introduction

A state of good mental health is not limited to the absence of mental illnesses, but is also described as a state of well-being in all bodily, psychological and social domains (Diener et al., 2009). The concept of subjective well-being (SWB) encompasses both negative (e.g., depression and anxiety) and positive aspects (e.g., happiness, satisfaction and contentment) (McDowell, 2010; Barden et al., 2015). The SWB construct is complex, as it concerns the cognitive, behavioral, emotional, social and personal spheres of human experience, and their optimal functioning (Keyes, 2002; Huppert and Ruggeri, 2018). SWB may have various connotations for different populations and cultures (World Health Organization, 2017). It has universally and consistently been proven to be a key outcome and predictor of several major life domains, and to contribute to both physical and mental health (Kansky, 2017). SWB has been found to be closely related to a range of important life domains, including positive development, successful learning (Diener et al., 2017), high-quality social relationships, better academic/work performance, less mental distress, and increased resilience in the face of stressors (Kansky, 2017). Given the well-established impact of SWB on health, several researchers have called for its inclusion as an outcome measure of mental health programs (Thornicroft and Slade, 2014). Therefore, several countries have already included SWB as a routine assessment to inform government decisions and public policy (Dolan et al., 2011; Helliwell et al., 2021). In recent years, particular emphasis has been placed on collecting self-rated SWB data in clinical settings (Topp et al., 2015), in the general population (De Kock et al., 2021), and in research (Topp et al., 2015; Lara-Cabrera et al., 2020) in an attempt to deepen understanding of the SWB concept and its applications.

One of the well-known, free-to-use and most widely used scales for assessing SWB is the WHO 5-item (WHO-5) Well-Being Index (World Health Organization, 1998; Topp et al., 2015). The WHO-5 Well-Being Index allows for a simple, brief self-report evaluation of the SWB construct over a 2-week period. It contains five positively worded items scored on a six-point scale. All items focus on positive health statements (Topp et al., 2015) and measure a global hedonic dimension of SWB (Bech, 2012). The WHO-5 Well-Being Index has demonstrated good psychometric qualities in a unidimensional structure, with high internal consistency and high convergent associations with other well-being measures (e.g., Bech et al., 2003). Since its development, the WHO-5 Well-Being Index has gained global popularity and has been translated into more than 30 languages (World Health Organization, 2024), predominantly in high-income Western and Asia-Pacific settings. The different linguistic versions of the WHO-5 Well-Being Index include Icelandic (Guðmundsdóttir et al., 2014), Swedish (Löve et al., 2014, Spanish (Bonnín et al., 2018), Polish (Cichoń et al., 2020), Italian (Nicolucci et al., 2004), Romanian (Preoteasa and Preoteasa, 2015), Danish (Schougaard et al., 2018), Sinhala (Perera et al., 2020), Brazilian Portuguese (de Souza and Hidalgo, 2012), Farsi (Dadfar et al., 2018), Turkish (Eser et al., 2019), Malay (Suhaimi et al., 2022), Thai (Saipanish et al., 2009), Taiwanese (Lin et al., 2013), Bangla (Faruk et al., 2021), Japanese (Awata et al., 2007), Korean (Moon et al., 2014), Chinese (Fung et al., 2022), and Swahili Kenyan (Chongwo et al., 2018). All these versions confirmed the robustness of the WHO-5 Well-Being Index and its utility in different research settings and across different geographical contexts (Topp et al., 2015). Over the years, the WHO-5 Well-Being Index has been increasingly and largely adopted for epidemiological research in various fields, including pediatrics (Allgaier et al., 2012),

adolescentology (Rose et al., 2017), geriatrics (Allgaier et al., 2013), occupational psychology (Sischka et al., 2018), and coronavirus disease 2019 (COVID-19)-related research (Lara-Cabrera et al., 2022). Furthermore, numerous studies have indicated that the WHO-5 Well-Being Index is suitable as a measure to screen for depression (Allgaier et al., 2013; Omani-Samani et al., 2019) and to monitor treatment response (Newnham et al., 2010a, 2010b).

We found three previous validations of the WHO-5 Well-Being Index in the Arabic language. The first one was performed in Lebanon among a relatively small sample ($N = 121$) and a gender-disproportionate group (75.2% females) composed of both community-dwelling and outpatient older individuals (Sibai et al., 2009). Results indicated that the Arabic WHO-5 Well-Being Index had satisfactory external and internal validity in detecting depression among Lebanese older adults (Sibai et al., 2009). The second validation was performed among a small sample of Saudi adults ($N = 190$, 59.5% females) and revealed a unidimensional latent structure of the scale, as well as high reliability and good convergent/divergent validity (Kassab Alshayea, 2023. The third validation was performed in a sample of patients with schizophrenia from Lebanon, in whom the WHO-5 Well-Being Index showed a unidimensional structure, good internal consistency reliability ($\alpha = .80$), cross-gender measurement invariance and good concurrent validity (Fekih-Romdhane et al., 2024).

### Well-being: Arab perspectives

People from Arab countries have been struggling over the past years with a high burden of mental health problems (Ibrahim, 2021). Mental disorder rates have exceeded the expected levels in Eastern Mediterranean Arab countries, resulting in a steadily increasing burden of disease (Mokdad et al., 2018). This burden is expected to rise due to the unstable political, economic and social climate in the Arab region (e.g., Charlson et al., 2012; Farran, 2021), and mental health will likely pose major challenges and strains on the already fragile resources in the coming years (Charara et al., 2017). Despite these alarming predictions, mental health care systems in Arab countries continue to be centralized, hospital-based and mainly focused on secondary care and disease treatment, thus neglecting the crucial role that SWB may play in alleviating mental health issues and promoting adaptive psychological outcomes (Basurrah et al., 2022). Such strategies are inappropriate and ineffective for dealing with mental health in the Arab population. Therefore, contextualized and culturally sensitive prevention approaches focused on SWB are urgently needed in Arab countries.

Recently, growing attention has been directed to the positive psychology field, and initial local research initiatives aiming at promoting SWB have begun to emerge (Basurrah et al., 2021). However, emerging studies are in no way comparable to non-Arab research in this field, both in terms of quality and quantity (Basurrah et al., 2021). In addition, experimental research on SWB in the Arab region is still in its infancy and suffers from major methodological flaws (Basurrah et al., 2021). We found only limited information available on SWB among Arab people, and very few studies using the WHO-5 Well-Being Index while focusing on specific populations (e.g., Youth in Jordan; Jamaluddine and Sieverding, 2022), Saudi women (Jradi and Abouabbas, 2017), Emirati and other Arabic-speaking adults (Elbarazi et al., 2022) and aid workers exposed to cumulative trauma in Palestine (Veronese et al., 2017). One of the main factors that hampers advances in mental health research and access to evidence-informed care in Arab

countries is the lack of valid and reliable measurement tools (Zeinoun et al., 2020). Providing psychometrically sound measures of the SWB construct could aid in designing and implementing evidence-informed interventions aimed at improving Arab people's well-being.

### Rationale of the present study

SWB is a culturally dependent and context-driven concept (Rice and Steele, 2004; Tov and Diener, 2009). There is evidence that individuals from collectivist cultures tend to exhibit lower ratings compared to those from individualistic cultures, which may result in distinct levels of functioning for the WHO-5 Well-Being Items (Brailovskaia et al., 2022). Despite this evidence, the cross-country validity of well-being scales remains underexplored (Cooke et al., 2016). The vast majority of previous validation and adaptation studies of the WHO-5 Well-Being Index were performed in Western countries with individualistic backgrounds (Zhang et al., 2024). In addition, the limited body of research available on the cross-country validity of the WHO-5 Well-Being Index has mainly involved Western and Asian countries. For example, Carrozzino et al. (2022) investigated the validity of the WHO-5 Well-Being Index in a sample of 3,762 adults from 5 European (i.e., Italy, Poland and Denmark) and non-European (i.e., China and Japan) countries. Sischka et al. (2020) demonstrated that the WHO-5 Well-Being Index is psychometrically appropriate and cross-nationally applicable in different nationally representative samples of individuals ($N = 43,469$) across 35 European countries. Another study also found that the WHO-5 Well-Being Index showed good validity and reliability across Spain, Chile and Norway in nurses who worked during the COVID-19 pandemic (Lara-Cabrera et al., 2022). More recently, a large multinational study confirmed the unidimensional measurement structure of the WHO-5 Well-Being Index in a sample of adolescents from 43 countries (in Europe, Central Asia and North America) (Sischka et al., 2025). The study also demonstrated configural and metric cross-country invariance, as well as appropriate patterns of correlations with life satisfaction, self-rated health, loneliness and psychosomatic complaints (Sischka et al., 2025). Cross-country validation studies are crucial to prove that the measure covers transcultural components of the subjective well-being construct, and can be used for cross-country comparison purposes in international multicenter research.

Although people from different Arab countries share similarities (including the language, geography, collectivist identity, religion and a young age structure; Harb, 2016), diversity also exists. Large cross-country studies have shown that the way Arab people view and behave toward mental health issues is not uniform and appears to be largely shaped by the local context of each Arab country (Fekih-Romdhane et al., 2023a, 2023b). Taking into consideration these cultural disparities, it is necessary to examine whether the WHO-5 Well-Being Index measures the SWB construct accurately in different Arab countries and cultural backgrounds. In this article, we aimed to contribute to the literature on SWB in different ways. First, we propose to investigate, for the first time, the cross-country validity of the WHO-5 Well-Being Index across different Arab countries to ensure its suitability for capturing and providing reliable information on the SWB construct in different Arab contexts. Second, as the two previous validations were conducted in Arab Middle East countries, we intended to expand our investigation to North African countries (i.e., Tunisia and Morocco) that have not been the subject of previous validation studies of the WHO-5 Well-Being Index. Third, we sought to examine

psychometric properties that have not been previously examined, such as measurement invariance across genders. Gender differences in SWB are culturally determined, as they may be substantially affected by social norms and adherence to traditional gender roles (Matud et al., 2019). However, variations across genders may also be largely driven by methodological factors (Graham and Chattopadhyay, 2013). For this reason, we sought to verify that the WHO-5 Well-Being Index invariantly measures the SWB factor across gender groups. Fourth, we aimed to explore its concurrent validity by calculating Pearson's correlation coefficients between the WHO-5 Well-Being Index and measures of depression, anxiety, stress, suicidal ideation and insomnia. We hypothesized that the Arabic version of the WHO-5 Well-Being Index would show a single-factor structure and satisfactory composite reliability in all samples from different countries, and would be invariant across gender groups. We also expected that the concurrent validity of the Arabic WHO-5 Well-Being Index would be supported through significant negative correlations with depression and other psychopathology measures.

## Methods

### Study design and participants

This was a multi-country, web-based, cross-sectional study. Several researchers from different institutions in the 22 Arab countries were invited to collaborate in our multinational project and join our team as co-investigators and co-authors. Researchers from six Arab countries accepted our invitation: Tunisia, Lebanon, Egypt, Jordan, Morocco and Kuwait. Arabic-speaking individuals from the general population, aged between 18 and 35 years and residing in an Arab country during the study period, were considered eligible to participate. This age range was chosen to guarantee homogeneous sampling and eliminate any differences resulting from age. Young adults aged 18–35 years have been found to display a worse health profile than both adolescents and those in their late 30s (Stroud et al., 2015). The committee on Improving the Health, Safety and Well-Being of Young Adults (convened by the National Research Council and the Institute of Medicine) concluded in their report that young adulthood is developmentally "of critical nature" within the life course (Committee on Improving the Health, Safety, and Well-Being of Young Adults et al., 2015). Accordingly, the committee recommended that "outcomes should be measured specifically for young adults," and that young adults should be treated as a distinct subpopulation in programming, planning, policy and research (Committee on Improving the Health, Safety, and Well-Being of Young Adults et al., 2015). Following these recommendations, we aimed to test the psychometric properties of the WHO-5 Well-Being Index exclusively in young adults within a relatively narrow age range. The survey was open between February and June 2022, and all responses collected during that period were included in the analysis.

All participants fulfilling these criteria were sampled using a convenience sampling technique and were invited to respond to a uniform, anonymous web-based questionnaire through social media platforms (including Instagram, Facebook and WhatsApp). Recruitment via Instagram was done using posts and stories shared by the research team and collaborators. The posts included a description of the study, the eligibility criteria and a link to the survey. No specific hashtags were used. Engagement was driven through reposts and snowball sampling. Participants were also asked to forward the link to other eligible people they might know,

using the snowball technique (Parker et al., 2019). Snowballing techniques and online recruitment of non-help-seeking participants are typically adopted for research in this area (e.g., see Preti et al., 2018). This recruitment approach was also chosen because several Arab countries boast high Internet penetration rates (varying from 72% in Tunisia to 100% in Kuwait) (World Bank, 2023), and some of the highest rates of social media usage in the world (Radcliffe et al., 2023). Eight out of 10 Arab youth aged 18–24 years reported daily usage of messaging apps, including Facebook (72%), Instagram (61%) and YouTube (53%) (Radcliffe et al., 2023). The questionnaire was administered using the free online survey tool provided by Google Forms. The study information and answering instructions were provided online via text; participants were asked to read them and give their informed consent before filling out the survey. Participants did not receive any incentives for participation. The study was performed in accordance with the Declaration of Helsinki for human research. The research protocol was approved by the Ethics Committees of the home institutions of the Principal Investigators, the Psychiatric Hospital of the Cross Ethics Committee, Jal Eddib, Lebanon (Ref: HPC-012-2022), and the Ethics Committee of the Razi Psychiatric Hospital, Manouba, Tunisia (Ref: ECRPH-2022-0019).

The total sample consisted of 3,247 participants, with a mean age of 23.36 ± 4.62 years. The majority of the participants were females (71.6%), single (75.9%) and had a university level of education (79.5%). The details of the sample by country are summarized in Supplementary Material (Supplementary Table S1).

### Minimum sample size

As a rule of thumb, simulation studies show that with normally distributed indicator variables and no missing data, a reasonable sample size for a simple Confirmatory Factor Analysis (CFA) model is about $N = 150$ (Muthén and Muthén, 2002), which was far exceeded in our sample. The resulting sample size was sufficiently large to provide adequate statistical power for all our analyses, including measurement invariance testing across groups (Meade et al., 2008).

### Measures

#### The WHO-5 Well-Being Index

This instrument was developed in 1998 and has been translated into 30 different languages. The WHO-5 consists of five items and assesses subjective psychological well-being. Each item is scored on a 5-point Likert scale with 5 = *all of the time* to 0 = *none of the time*. Therefore, the total score ranges from 0 (*absence of well-being*) to 25 (*maximum well-being*) (World Health Organization 2019). Raw scores are then multiplied by 4 to obtain a percentage score ranging from 0 (worst) to 100 (best). The Arabic version of this instrument was validated in Lebanon among elderly people (Sibai et al., 2009).

#### Columbia-Suicide Severity Rating Scale (C-SSRS)

This scale is composed of five items, rated as a no/yes type of answer. It evaluates suicidality over the past month. Higher scores indicate higher suicidal ideation. This scale has been validated in the Arabic language among Arabic-speaking adults from Lebanon, where it showed a unidimensional factor structure, good internal consistency ($\alpha = .797$) and appropriate convergent validity with measures of depression, anxiety and self-esteem (Zakhour et al.,

2021). In the present sample, the C-SSRS yielded a McDonald's $\omega$ of .79 and a Cronbach's $\alpha$ of .79.

#### Insomnia Severity Index (ISI)

This scale is composed of seven items, rated on a 4-point Likert scale. Higher scores reflect more severe insomnia. The Arabic validated version of the ISI was used, which demonstrated good reliability ($\alpha = .833$) and good validity in a sample of Arabic-speaking community-dwelling adults from Lebanon (Hallit et al., 2019). In the present sample, the ISI yielded a McDonald's $\omega$ of .82 and a Cronbach's $\alpha$ of .82.

#### Depression, Anxiety, and Stress Scale 8 items (DASS-8)

The DASS-8 is composed of eight items measuring depression (three items), anxiety (three items) and stress (two items). Items are rated on a 4-point Likert scale. Higher scores reflect higher depression, anxiety and stress. The Arabic-validated DASS-8 was used in this study, which showed excellent psychometric properties in terms of internal consistency ($\alpha = .94$), convergent validity, predictive validity and discriminant validity (Ali et al., 2024). In this study, the DASS-8 showed good internal consistency reliability for all three dimensions: depression ($\omega = .91/\alpha = .91$), anxiety ($\omega = .90/\alpha = .90$) and stress ($\omega = .73/\alpha = .73$).

#### Demographics

Participants were asked to provide their demographic details, including age, gender and education level.

### Analytic strategy

#### Confirmatory Factor Analysis

There were no missing responses in the dataset since all questions were required in the Google Forms. Duplicate responses were screened and removed using Excel's "Remove Duplicates" function, based on identical patterns in response time stamps and item-level data. We used data from the total sample to conduct a CFA using the SPSS AMOS v.29 software. Our intention was to test the original model of the WHO-5 Well-being scale (i.e., one-factor model). Parameter estimates were obtained using the maximum likelihood method with corresponding fit indices. To identify the model, we used the marker variable approach (Little et al., 2002), in which the factor loading of the first item (Well-being 1) was fixed to 1 to scale the latent variable. This is a common method for setting the metric of latent constructs in CFA (Schreiber, 2008, 2017). Multiple indices were calculated to assess model fit: the normed model chi-square ($\chi^2$/df), the root mean square error of approximation (RMSEA), the Tucker–Lewis Index (TLI) and the comparative fit index (CFI). Values ≤5 for $\chi^2$/df, ≤0.08 for RMSEA and 0.90 for CFI and TLI indicate good fit of the model to the data (Hu and Bentler, 1999). In addition to reporting global fit indices (RMSEA, CFI, TLI and standardized root mean square residual [SRMR]), we examined local fit through standardized residual covariances and modification indices, as recommended by previous authors (Steiger, 2007; Kline, 2023; Goretzko et al., 2024). These local diagnostics help identify specific areas of model misfit that global indices may obscure. Moreover, evidence of convergent validity was assessed in this subsample using the Fornell–Larcker criterion, with average variance extracted (AVE) values of ≥0.50 considered adequate (Malhotra and Dash, 2011). Multivariate normality was not verified at first (Bollen-Stine bootstrap $p = .002$); therefore, we performed a nonparametric bootstrapping procedure.

## Measurement invariance

To examine gender, country and age (dichotomized into ≤25 vs. ≥26 years (Carlucci et al., 2018) invariance of WHO-5 Well-being scores, we conducted multigroup CFA (Chen, 2007) using the total sample. Measurement invariance was assessed at the configural, metric and scalar levels (Vadenberg and Lance, 2000). We accepted ΔCFI ≤ 0.010 and ΔRMSEA ≤ 0.015 or ΔSRMR ≤ 0.010 as evidence of invariance (Chen, 2007). Differences between genders and age groups were evaluated using the Student $t$-test, and differences between countries were evaluated using the analysis of variance test.

## Further analyses

Reliability was assessed using McDonald's $\omega$ and Cronbach's $\alpha$, with values >.70 reflecting adequate reliability (Malkewitz et al., 2023). The WHO-5 Well-being total score was considered normally distributed since the skewness and kurtosis values fell between ±1 (Hair et al., 2017). Therefore, to assess concurrent and divergent validity, we examined bivariate correlations between the WHO-5 Well-Being Index and the CSRS, ISI and DASS-8 scores using the Pearson test. Based on Cohen (1992), values ≤.10 were considered weak, ~.30 as moderate and ~.50 as strong correlations.

## Results

The information on the distribution (mean, SD, skewness and kurtosis) of each WHO-5 Well-Being Index item, stratified by country and gender, is summarized in Table 1.

### CFA of the Arabic WHO-5 Well-Being Index

Except for the RMSEA, most CFA model fit indices indicated that the fit of the one-factor model of the Arabic WHO-5 Well-Being Index was acceptable: $\chi^2$ = 223.44, df = 5 ($p < .001$), RMSEA = 0.116 (90% confidence interval [CI] = 0.103, 0.129), SRMR = 0.017, CFI = 0.985, TLI = 0.970. When a correlation between the residuals of items 1 and 4 was added (after showing a high modification index), the results improved further as follows: $\chi^2$ = 57.71, df = 4 ($p < .001$), RMSEA = 0.064 (90% CI = 0.050, 0.080), SRMR = 0.009, CFI = 0.996, TLI = 0.991. The standardized estimates of factor loadings (Figure 1) and the AVE values (0.77) were all excellent. The same analysis was conducted for each country and showed adequate results as well (Table 2). The results of the standardized residual covariances and modification indices can be found in Supplementary Tables S2 and S3.

### Internal and composite reliability

Internal reliability of the WHO-5 Well-being scores was adequate in the total sample ($\omega$ = .94/$\alpha$ = .94), in men ($\omega$ = .95/$\alpha$ = .95) and in women ($\omega$ = .94/$\alpha$ = .94), in participants aged ≤25 years ($\omega$ = .94/$\alpha$ = .94) and ≥ 26 years ($\omega$ = .96/$\alpha$ = .96) and within each country as follows: Tunisia ($\omega$ = .96/$\alpha$ = .96), Lebanon ($\omega$ = .95/$\alpha$ = .96), Kuwait ($\omega$ = .94/$\alpha$ = .94), Egypt ($\omega$ = .92/$\alpha$ = .92), Jordan ($\omega$ = .93/$\alpha$ = .93) and Morocco ($\omega$ = .94/$\alpha$ = .94).

### Measurement invariance and differences by gender, age and country

As reported in Table 3, indices suggested that configural, metric and scalar invariance were supported across gender, country and age categories. The results showed that a significantly higher mean WHO-5 Well-being score was found in males compared to females (10.30 ± 6.65 vs. 8.73 ± 6.17; $t$ (3245) = 6.37; $p < .001$, Cohen's $d$ = 0.248). Moreover, the highest mean well-being score was found in Morocco (44.2 ± 26.84) and Kuwait (40.4 ± 27.04), followed by Lebanon (39.8 ± 25.24), Jordan (36.84 ± 24.04), Tunisia (33.6 ± 25.88) and Egypt (32.2 ± 22.72), with the difference being significant ($F$ = 15.96, $p < .001$). The Bonferroni post-hoc analysis showed a significant difference between Tunisia and Lebanon ($p < .001$), Tunisia and Kuwait ($p < .001$), Tunisia and Morocco ($p < .001$), Lebanon and Egypt ($p < .001$), Kuwait and Egypt ($p < .001$), Egypt and Morocco ($p < .001$), and Jordan and Morocco ($p = .027$).

Finally, a higher mean well-being score was found in participants aged 26 years and above compared to those aged 25 years and below (9.61 ± 6.61 vs. 9.02 ± 6.25; $t$ (3,245) = −2.26; $p < .001$, Cohen's $d$ = .093).

### Concurrent validity (total sample)

As for concurrent validity, WHO-5 Well-being scores showed a moderate significant inverse correlation with DASS depression ($r = −.28$; $p < .001$), anxiety ($r = −.29$; $p < .001$) and stress ($r = −.27$; $p < .001$) subscales scores, suicidal ideation ($r = −.16$; $p < .001$) and insomnia severity ($r = −.37$; $p < .001$) (Table 4).

## Discussion

This study is the first to explore the cross-country validity of the WHO-5 Well-Being Index among young adults across six Arab countries (i.e., Tunisia, Lebanon, Egypt, Jordan, Morocco and Kuwait) in the Middle East and North Africa region. Results showed that all five items loaded onto a single latent factor in both genders and across all six countries, demonstrating adequate reliability, as well as good convergent and divergent validity. Overall, these findings suggest that the Arabic WHO-5 measures the originally intended SWB construct within the specific context conditions of Arabic-speaking populations. By verifying the cross-country validity of the Arabic WHO-5, our study supports its wider application to epidemiologically explore SWB among Arabic-speaking young adults from broad geographic areas.

We found that WHO-5 mean scores varied significantly across countries, ranging from 32.2 ± 22.72 in Egypt to 44.2 ± 26.84 in Morocco. Despite these wide variations, WHO-5 scores reported in all six Arab countries were much lower than those observed among the adult general population in other international studies (e.g., 56 in Sri Lanka [Perera et al., 2020], 64.74 in Iceland [Guðmundsdóttir et al., 2014], 73.37 in southern Brazil [de Souza and Hidalgo, 2012], but were comparable to scores found in a Middle Eastern country (i.e., 35.8 in Iranian people [Dadfar et al., 2018]). It should be noted that direct comparisons of WHO-5 Well-being mean scores between studies may not be meaningful due to differences in contextual factors, such as demographics (e.g., our sample exclusively included young adults). That said, the low mean scores observed in our present sample should serve as a warning for clinicians, researchers and policy-makers working in Arab settings, and further highlight that local culturally sensitive strategies are needed to address well-being issues among Arab young adults.

The construct validity of the WHO-5 was examined using CFA, which is consistently advocated by validation researchers as a crucial step in scale validation (Loewenthal and Lewis, 2018;

**Table 1.** Distribution (mean, SD, skewness and kurtosis) of each WHO-5 Well-Being Index item, stratified by country and gender

|  | Mean | SD | Skewness | Kurtosis |
|---|---|---|---|---|
| **(a) Total sample** | | | | |
| Well-being 1 | 2.04 | 1.42 | 0.314 | −0.875 |
| Well-being 2 | 1.80 | 1.37 | 0.485 | −0.625 |
| Well-being 3 | 1.84 | 1.39 | 0.460 | −0.677 |
| Well-being 4 | 1.70 | 1.41 | 0.577 | −0.554 |
| Well-being 5 | 1.79 | 1.46 | 0.551 | −0.688 |
| **(b) Tunisia** | | | | |
| Well-being 1 | 1.72 | 1.41 | 0.700 | −0.400 |
| Well-being 2 | 1.64 | 1.33 | 0.742 | −0.246 |
| Well-being 3 | 1.73 | 1.38 | 0.650 | −0.420 |
| Well-being 4 | 1.67 | 1.41 | 0.728 | −0.348 |
| Well-being 5 | 1.64 | 1.42 | 0.788 | −0.314 |
| **(c) Lebanon** | | | | |
| Well-being 1 | 2.20 | 1.37 | 0.137 | −0.918 |
| Well-being 2 | 1.93 | 1.36 | 0.293 | −0.808 |
| Well-being 3 | 1.98 | 1.35 | 0.306 | −0.769 |
| Well-being 4 | 1.84 | 1.40 | 0.352 | −0.763 |
| Well-being 5 | 2.00 | 1.42 | 0.349 | −0.887 |
| **(d) Kuwait** | | | | |
| Well-being 1 | 2.27 | 1.53 | 0.129 | −1.07 |
| Well-being 2 | 2.01 | 1.49 | 0.354 | −0.900 |
| Well-being 3 | 1.97 | 1.49 | 0.361 | −0.888 |
| Well-being 4 | 1.86 | 1.53 | 0.472 | −0.825 |
| Well-being 5 | 1.99 | 1.55 | 0.377 | −0.939 |
| **(e) Egypt** | | | | |
| Well-being 1 | 1.84 | 1.31 | 0.409 | −0.738 |
| Well-being 2 | 1.60 | 1.27 | 0.584 | −0.321 |
| Well-being 3 | 1.63 | 1.31 | 0.568 | −0.463 |
| Well-being 4 | 1.45 | 1.27 | 0.702 | −0.243 |
| Well-being 5 | 1.54 | 1.38 | 0.731 | −0.333 |
| **(f) Jordan** | | | | |
| Well-being 1 | 2.15 | 1.36 | 0.294 | −0.628 |
| Well-being 2 | 1.81 | 1.29 | 0.370 | −0.576 |
| Well-being 3 | 1.82 | 1.34 | 0.420 | −0.510 |
| Well-being 4 | 1.71 | 1.41 | 0.561 | −0.456 |
| Well-being 5 | 1.72 | 1.38 | 0.610 | −0.320 |
| **(g) Morocco** | | | | |
| Well-being 1 | 2.58 | 1.42 | −0.167 | −0.904 |
| Well-being 2 | 2.11 | 1.47 | 0.207 | −0.987 |
| Well-being 3 | 2.24 | 1.48 | 0.147 | −1.032 |
| Well-being 4 | 1.99 | 1.53 | 0.354 | −0.908 |
| Well-being 5 | 2.13 | 1.54 | 0.232 | −1.126 |

*(Continued)*

**Table 1.** *(Continued)*

|  | Mean | SD | Skewness | Kurtosis |
|---|---|---|---|---|
| **(h) Males** | | | | |
| Well-being 1 | 2.19 | 1.47 | 0.242 | −0.962 |
| Well-being 2 | 2.06 | 1.42 | 0.353 | −0.795 |
| Well-being 3 | 2.06 | 1.43 | 0.353 | −0.823 |
| Well-being 4 | 1.97 | 1.48 | 0.428 | −0.785 |
| Well-being 5 | 2.02 | 1.50 | 0.405 | −0.875 |
| **(i) Females** | | | | |
| Well-being 1 | 1.99 | 1.40 | 0.336 | −0.842 |
| Well-being 2 | 1.70 | 1.34 | 0.534 | −0.544 |
| Well-being 3 | 1.75 | 1.37 | 0.499 | −0.612 |
| Well-being 4 | 1.60 | 1.37 | 0.630 | −0.449 |
| Well-being 5 | 1.70 | 1.43 | 0.609 | −0.592 |

Zeng et al., 2020). Unlike exploratory factor analysis (EFA), CFA imposes meaningful constraints when evaluating a measure's validity (Guðmundsdóttir et al., 2014). Although the WHO-5 has been extensively validated in dozens of languages and countries, few WHO-5 Well-being assessments have used CFA (De Wit et al., 2007; Fung et al., 2022), and several validation studies have relied only on EFA (Allgaier et al., 2012; Awata et al., 2007; Bonnín et al., 2018; Cichoń et al., 2020; Hochberg et al., 2012; Löve et al., 2014). Analyses from the present study showed that the fit of a one-factor model to the data was acceptable in each of the six countries, thus replicating the factor structure of the original WHO-5 Well-Being Index (Bech, 2004, 2012; Bech et al., 2003), and that obtained in other linguistic versions using CFA (e.g., Swahili Kenyan [Chongwo et al., 2018], Malay [Suhaimi et al., 2022], Icelandic [Guðmundsdóttir et al., 2014], Sinhala [Perera et al., 2020], Chinese [Fung et al., 2022] and Arabic [Kassab Alshayea, 2023]). Our results support the applicability of the WHO-5 Well-Being Index as a unidimensional measure of SWB in Arab contexts. Furthermore, to assess composite reliability in our sample, McDonalds' $\omega$ coefficients were used as they have been shown to provide more realistic estimates of a measure's reliability than Cronbach's $\alpha$ (Ravinder and Saraswathi, 2020). Findings revealed that the Arabic WHO-5 Well-Being Index yielded high reliability coefficients in the total sample and both genders, which is in line with previous international studies on other translations of the WHO-5 Well-Being Index that mostly relied on Cronbach's $\alpha$ coefficients (e.g., $\alpha = 0.79$–0.91 in Italy [Nicolucci et al., 2004], 0.86–0.88 in Kenya [Chongwo et al., 2018], 0.87 in Poland [Cichoń et al., 2020], 0.88 in Romania [Preoteasa and Preoteasa, 2015], 0.83 in Sweden [Löve et al., 2014], 0.91 in Iran [Dadfar et al., 2018], 0.81 in Turkey [Eser et al., 2019], 0.91 in Malaysia [Suhaimi et al., 2022], 0.75 in Bangladesh [Faruk et al., 2021], 0.83 in Brazil [de Souza and Hidalgo, 2012], 0.82–0.87 in Iceland [Guðmundsdóttir et al., 2014], 0.81–0.85 in China [Fung et al., 2022], 0.91 in Saudi Arabia [Kassab Alshayea, 2023]). In the initial unidimensional CFA model, global model fit was adequate but could be improved. Based on the modification indices and in line with prior findings (Goretzko et al., 2024), we allowed residuals of items 1 and 4 to correlate since these items reflect affective well-being and share similar emotional tone and response patterns, which may explain the residual association beyond the general

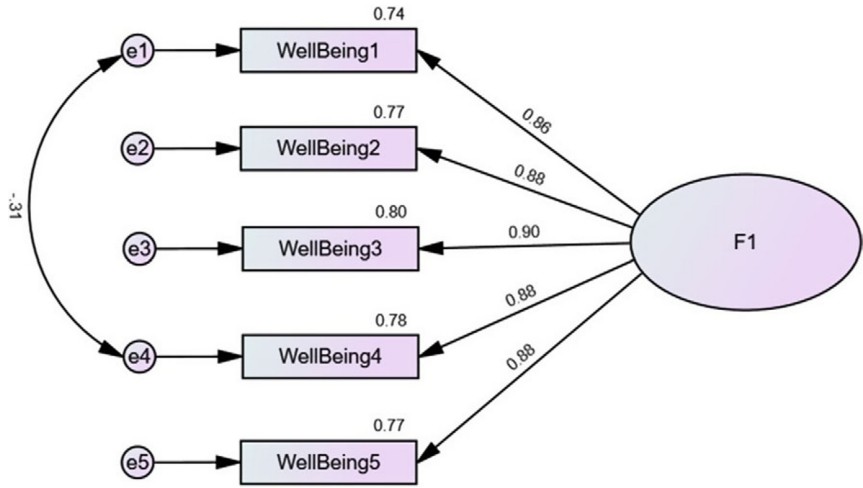

**Figure 1.** Standardized factor loadings derived from the Confirmatory Factor Analysis of the Arabic WHO-5 Well-Being Index in the total sample.

**Table 2.** Confirmatory Factor Analysis and standardized loading factors of the Arabic WHO-5's items per country

|  | Tunisia | Lebanon | Kuwait | Egypt | Jordan | Morocco |
|---|---|---|---|---|---|---|
| SRMR | 0.011 | 0.008 | 0.013 | 0.010 | 0.016 | 0.007 |
| $\chi^2$ | 34.73 | 13.63 | 20.55 | 13.40 | 15.82 | 2.47 |
| df | 4 | 4 | 4 | 4 | 4 | 4 |
| TLI | 0.980 | 0.992 | 0.985 | 0.992 | 0.971 | 1.004 |
| CFI | 0.992 | 0.997 | 0.994 | 0.997 | 0.989 | 1.000 |
| RMSEA [90% CI] | 0.112 [0.079–0.147] | 0.063 [0.028–0.101] | 0.079 [0.047–0.114] | 0.051 [0.023–0.083] | 0.107 [0.055–0.165] | 0.001 [<0.001–0.085] |
| **Standardized loading factors** | | | | | | |
| Item 1 | 0.88 | 0.87 | 0.87 | 0.84 | 0.82 | 0.83 |
| Item 2 | 0.92 | 0.90 | 0.86 | 0.83 | 0.88 | 0.87 |
| Item 3 | 0.93 | 0.90 | 0.87 | 0.83 | 0.84 | 0.89 |
| Item 4 | 0.94 | 0.90 | 0.86 | 0.85 | 0.86 | 0.88 |
| Item 5 | 0.93 | 0.92 | 0.89 | 0.85 | 0.89 | 0.93 |

well-being factor. This correlation suggests that these items may consistently share unexplained variance. This modification improved model fit substantially, while remaining theoretically justified.

For Morocco, the fit indices showed unusually high values (TLI = 1.004, CFI = 1.000, RMSEA CI < 0.001–0.085), suggesting potential overfitting, model saturation, limited variance or a small sample size; these conditions can inflate fit indices and underestimate model misfit. In this study, this can be explained by the small sample from Morocco (*n* = 202), which is supported in the literature (Xia and Yang, 2019). For Tunisia and Jordan, the RMSEA values were beyond the acceptable limits, indicating poor fit, although CFI and TLI suggested otherwise. This should be interpreted with caution, as RMSEA is sensitive to low degrees of freedom and can falsely indicate poor fit in small or simple models (Kenny et al., 2015).

Another relevant contribution of this study was to examine the measurement invariance of the WHO-5 Well-being scores across gender and countries. Results from multigroup analyses demonstrated that configural, metric and scalar invariance were supported across gender in the total sample and by country. Evidence of invariance across gender groups has also been reported in other validation studies and different linguistic contexts (e.g., Icelandic national and patient samples [Guðmundsdóttir et al., 2014] and Sri Lankan people from the general population [Perera et al., 2020]). These findings imply that WHO-5 Well-being mean differences in SWB between male and female respondents, as well as between respondents from various Arab countries, are not attributable to group-level variations in understanding or responding to items, but to real differences in the construct level (Putnick and Bornstein, 2016). We, therefore, suggest that the Arabic WHO-5 Well-Being Index can be used reliably to compare mean differences between gender and country groups.

In this regard, we found that males displayed higher WHO-5 Well-being scores than females, which is in accordance with previous studies (e.g., Nicolucci et al., 2004; Lin et al., 2013; Guðmundsdóttir et al., 2014; Löve et al., 2014; Preoteasa and Preoteasa, 2015). Furthermore, the highest well-being scores were

**Table 3.** Measurement invariance of the Arabic WHO-5 Well-Being Index in the total sample

| Model | $\chi^2$(df) | p | CFI | RMSEA | SRMR | Model comparison | $\Delta\chi^2$ | $\Delta$CFI | $\Delta$RMSEA | $\Delta$SRMR | $\Delta$df | p |
|---|---|---|---|---|---|---|---|---|---|---|---|---|
| **Model 1: Invariance by gender** | | | | | | | | | | | | |
| Males | 31.54 (4) | <.001 | 0.994 | 0.087 | 0.010 | | | | | | | |
| Females | 44.98 (4) | <.001 | 0.996 | 0.066 | 0.010 | | | | | | | |
| Configural | 242.89 (10) | <.001 | 0.984 | 0.085 | 0.018 | | | | | | | |
| Metric | 247.73 (14) | <.001 | 0.984 | 0.072 | 0.018 | Configural vs. metric | 4.84 | <0.001 | 0.013 | <0.001 | 4 | .304 |
| Scalar | 268.75 (18) | <.001 | 0.983 | 0.066 | 0.018 | Metric vs. scalar | 21.02 | 0.001 | 0.006 | <0.001 | 4 | <.001 |
| **Model 2: Invariance by country** | | | | | | | | | | | | |
| Configural | 100.60 (24) | <.001 | 0.995 | 0.031 | 0.011 | | | | | | | |
| Metric | 116.69 (44) | <.001 | 0.995 | 0.023 | 0.011 | Configural vs. metric | 16.09 | <0.001 | 0.008 | <0.001 | 20 | .711 |
| Scalar | 211.45 (64) | <.001 | 0.990 | 0.027 | 0.012 | Metric vs. scalar | 94.76 | 0.005 | 0.004 | 0.001 | 20 | <.001 |
| **Model 3: Invariance by age** | | | | | | | | | | | | |
| Configural | 64.55 (8) | <.001 | 0.996 | 0.047 | 0.010 | | | | | | | |
| Metric | 70.41 (12) | <.001 | 0.996 | 0.039 | 0.010 | Configural vs. metric | 5.86 | <0.001 | 0.008 | <0.001 | 4 | .209 |
| Scalar | 87.90 (16) | <.001 | 0.995 | 0.037 | 0.010 | Metric vs. scalar | 17.49 | 0.001 | 0.002 | <0.001 | 4 | .001 |

*Note*: CFI, Comparative Fit Index; RMSEA, root mean square error of approximation; SRMR, standardized root mean square residual.

**Table 4.** Pearson correlation matrix

| | 1 | 2 | 3 | 4 | 5 |
|---|---|---|---|---|---|
| 1. Well-being | | | | | |
| 2. Depression | −.28*** | | | | |
| 3. Anxiety | −.29*** | .79*** | | | |
| 4. Stress | −.27*** | .76*** | .72*** | | |
| 5. Suicidal ideation | −.16*** | .18*** | .19*** | .08*** | |
| 6. Insomnia severity | −.37*** | .23*** | .25*** | .20*** | .33*** |

***p < .001.

the correlations between the WHO-5 Well-Being Index and different measures of mental health problems (e.g., anxiety [Awata et al., 2007; Guðmundsdóttir et al., 2014], stress [Guðmundsdóttir et al., 2014; Faruk et al., 2021] and sleep problems [Löve et al., 2014]). In sum, the current results strongly support the validity of the Arabic WHO-5 Well-Being Index and offer additional confirmation that it serves the purposes for which it was originally developed.

## Study limitations and research perspectives

Despite its valuable contribution to the field of SWB, the present study has some limitations that need to be addressed in future research. First, we did not use a structured clinical interview against which the results from the self-report measure could be validated, which prevented us from assessing the specificity and sensitivity of the WHO-5 Well-Being Index as a depression screening tool. To address this limitation, future studies should include an external criterion measure. Second, our data were gathered at a single point in time, which precluded us from testing the stability and invariance of the Arabic WHO-5 Well-Being Index over time. Therefore, additional validation studies are needed to examine the test–retest reliability of the scale. Additionally, because of the cross-sectional design, the relationships examined in this study cannot be interpreted causally. Third, the use of a community sample of young adults may undermine the generalizability of our findings to clinical populations. Fourth, we employed an online survey and convenience sampling, both of which mostly attracted highly educated and female participants, thereby limiting the representativeness of our sample. Fifth, the theoretically related measures used for validity purposes (C-SSRS, ISI and DASS-8) have not been validated in the different cultural contexts and countries involved in this study. Finally, the six countries involved in our study are lower-to-middle-income countries, and cannot be considered representative of all Arab populations and the Arab world. Young adults from low-income Arab countries (such as Yemen and Syria) may have different SWB levels, and should be the subject of future validation studies.

exhibited by Moroccan and Kuwaiti participants compared to those from other nationalities. Comparative studies on mental health and well-being between the different Arab countries are lacking. A study published in 2012 reported that Morocco has one of the lowest numbers of psychiatrists and higher prevalence rates of mental health problems compared to other Arab countries (Okasha et al., 2012). One plausible explanation for our findings could be the sociopolitical unrest that took place in some Arab countries over the past years, such as the Arab spring in Tunisia and Egypt, or the economic crises and conflicts in Lebanon (Al-ghzawi et al., 2014).

As for concurrent validity, WHO-5 Well-being scores showed a strong, significant inverse correlation with DASS-8 depression subscores, which is consistent with several previous validations using various depression measures (e.g., Cichoń et al., 2020; Dadfar et al., 2018; Guðmundsdóttir et al., 2014; Lucas-Carrasco, 2012; Perera et al., 2020; Saipanish et al., 2009). Similarly, the WHO-5 Well-Being Index has consistently shown high sensitivity and specificity in detecting depression, and has been extensively applied as a screening tool for this condition (Topp et al., 2015). Negative correlations have also been demonstrated between WHO-5 Well-being scores and different symptoms of mental health problems (anxiety, stress, suicidal ideation and insomnia) in our sample. These findings also concur with prior research comparing

Using dynamic fit index thresholds for model evaluation, reporting effect sizes for measurement non-invariance, and examining group mean differences (e.g., by gender, age or country) within a latent variable framework may also be valuable for future work.

## Conclusion

In summary, findings indicate that the WHO-5 Well-Being Index in its Arabic version has a unidimensional structure among Arabic-speaking young adults across six Arab countries, high internal consistency, good concurrent validity and measurement invariance across gender. This study contributes to the growing research in the field of positive psychology and well-being by providing a brief, valid and reliable Arabic version of the WHO-5 Well-Being Index that can be used cross-nationally with a variety of Arabic-speaking young adult populations for screening and research purposes.

**Open peer review.** To view the open peer review materials for this article, please visit http://doi.org/10.1017/gmh.2025.10051.

**Supplementary material.** The supplementary material for this article can be found at http://doi.org/10.1017/gmh.2025.10051.

**Data availability statement.** All data generated or analyzed during this study are not publicly available due to the restrictions from the ethics committee, but are available upon a reasonable request from the corresponding authors (FFR and SH).

**Acknowledgments.** The authors would like to thank all participants.

**Author contribution.** FFR and SH designed the study. FFR and SH wrote the article. SH carried out the analysis and interpreted the results. WC, AA, MF, HAMS, MH, IHMH, AYN, BZ, MC, YE-F, GY, GS, AH-M, ER, AH and SO were involved in the data collection. MC reviewed the article for intellectual content. All authors read and approved the final manuscript.

**Competing interests.** The authors declare none.

**Ethics approval and consent to participate.** Ethics approval for this study was obtained from the Psychiatric Hospital of the Cross ethics committee, Lebanon (Ref: HPC-012-2022). Written informed consent was obtained from all subjects; the online submission of the soft copy was considered equivalent to receiving written informed consent. All methods were performed in accordance with the relevant guidelines and regulations.

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
