## [Reviewer Report]

Thank you for the opportunity to review this manuscript, which investigates the cross-cultural psychometric properties (i.e., factor structure, internal consistency, measurement invariance across gender and concurrent validity) of an Arabic version of the WHO-5 Well-Being Index among non-clinical young adults from six Arab countries (i.e., Tunisia, Lebanon, Egypt, Jordan, Morocco, and Kuwait). Given that the WHO-5 is one of the most widely used and validated measures of well-being globally, and that the majority of psychometric research on this scale has been conducted in WEIRD (Western, Educated, Industrialized, Rich, and Democratic) populations (e.g., Topp et al., 2015; Sischka et al., 2025), the current study is timely and relevant. By focusing on young adults in Arab countries, the manuscript addresses an important gap in the cross-cultural generalizability of the WHO-5 and contributes to the growing literature on mental health measurement in non-WEIRD contexts. However, there are several conceptual, methodological, and reporting issues that should be addressed. Please see my detailed comments below.

Abstract

- You wrote: “We found that WHO-5 mean scores varied significantly across countries, ranging from 32.2 ± 22.72 in Egypt to 44.2 ± 26.84 in Morocco.” I assume you are referring to WHO-5 percentage scores, as standard mean scores would range between 0 and 5.

- Consider including McDonald’s omega values for both genders in the abstract to provide a quick impression of internal consistency.

- You might also report the actual correlation coefficients, rather than only describing the direction of the relationships, to give readers a clearer understanding of effect sizes.

Introduction

- I would suggest softening the following sentence: “Despite these data, the cross-cultural validity of well-being scales is still an unexplored question,” as many studies have already examined the cross-cultural validity and comparability of such scales—as you indicate in the following sentence. Consider rephrasing it to: “Despite this growing body of research, the cross-cultural validity of well-being scales remains underexplored.”.

- You might also want to discuss Sischka et al. (2025), who investigated the psychometric properties of the WHO-5 among adolescent populations across 43 countries (mainly in Europe, but also in Central Asia and North America). They reported correlations with life satisfaction, self-rated health, psychosomatic complaints, and loneliness. They also investigated measurement invariance by gender (see note 10). They found that the WHO-5 was largely measurement invariant across gender within each country.

- You wrote: “Large cross-cultural studies have shown that the way Arab people view and behave towards mental health issues is not uniform, and appears to be largely shaped by the local context of each Arab country.” This raises the possibility that the WHO-5 may not be measurement invariant across Arab countries. If so, I wonder why you did not examine cross-country measurement invariance in your own study?

Methods

- There are various methods for determining sample size (e.g., power analysis, precision analysis, sequential analysis, or practical constraints such as time and budget; see Giner-Sorolla et al., 2024). Please briefly explain how your sample size was determined.

- For the DASS-8, please specify how many items are included in each subscale.

- You stated that there were no missing responses in the dataset. This is somewhat unusual and may suggest that participants with missing data were removed (which is not the same as “no missing data”) or that a forced-response format was used (e.g., Sischka et al., 2022). Please clarify.

- Based on Figure 1, it appears that you used the “marker variable” approach (Little et al., 2006), likely using the first item as the marker. However, the method for scaling the latent variable should already be described in the “Analytical Strategy” section (e.g., Nye, 2023; Schreiber, 2008, 2017).

- Since global fit indices aggregate many discrepancies into a single value (Steiger, 2007), they may obscure local areas of poor model fit. Therefore, consider reporting local fit indices alongside global statistics (e.g., Goretzko et al., 2023; Kline, 2024). This would also allow comparisons with prior psychometric studies on the WHO-5, many of which have reported strongly correlated residuals between items 1 and 2. You could include this information in an online supplement.

- Would it be possible to calculate and report effect size indices for measurement invariance (e.g., Nye et al., 2019; Gun et al., 2020)?

- Relying on fixed cutoff values to judge model fit has been widely criticized, as these indices are influenced by factors unrelated to true model fit (e.g., estimator, number of response categories, item distributions; see Groskurth et al., 2024; Marsh et al., 2004; McNeish & Wolf, 2023). You might consider using dynamic fit index cutoffs, which can be easily calculated using this Shiny app: https://www.dynamicfit.app (McNeish & Wolf, 2023; McNeish, 2023).

- You wrote: “The absence of multicollinearity was verified through tolerance values > .2 and variance inflation factor (VIF) values < 5.” For which of your analyses was multicollinearity a concern? Please clarify the purpose of this check.

Results

- I recommend moving the information about sample size and the distribution of demographic variables to the Methods section under “Study Design and Participants.”

- Could you provide information on the distribution (mean, SD, skewness, kurtosis) of each WHO-5 item, stratified by country and gender?

- I am not aware of any reference—including Hu & Bentler (1999), which you cited—that considers RMSEA values above .100 as indicative of acceptable model fit. In fact, a RMSEA in this range is typically interpreted as indicating poor fit (e.g., Little, 2024, p. 143). While the model may still be “good enough” in some practical contexts, this should be evaluated through local fit statistics. Interestingly, this pattern of model fit indices (i.e., SRMR, CFI, TLI suggesting good fit, but RMSEA indicating poor fit) has also been observed in other cross-national studies (e.g., Sischka et al., 2020; 2025). Still, local model fit diagnostics are essential to determine what might be driving the discrepancy.

- For the country-stratified CFA results, please also report the p-value for the model chi-square test (see Kline, 2023, Chapter 10; Nye, 2023). Also, it is somewhat inconsistent to report CFAs separately by country while not conducting cross-country measurement invariance testing.

- In Table 2, please also include model fit statistics for the gender-stratified CFA results, as you did for the country-stratified results in Table 1.

- Did you compute composite reliability for the model in which the residual correlation between items 1 and 4 was freely estimated, or for the basic unidimensional model where all residuals were constrained to zero? Please clarify.

- Table 3 does not provide information on gender-based measurement invariance; it only presents mean and SD differences across countries. Therefore, Table 3 should not be referenced within the sentence discussing measurement invariance.

- Since Section 3.3 includes not only gender measurement invariance but also gender differences, the heading should be updated to reflect both.

- Please include a table for the correlations reported in Section 3.4. This would allow readers to assess the associations among all variables (e.g., between stress and anxiety), not just those described in the text.

Discussion

- You should also discuss that the retained model freely estimated the correlation between the residuals of items 1 and 4, indicating that a purely unidimensional model was not supported.

- Please note in the Limitations section that the associations identified cannot be interpreted causally. For further guidance on how to report limitations, see Clarke et al. (2024).

- Assessing the WHO-5 at multiple time points would also allow for testing its temporal invariance.

Minor points

- You switch between composite reliability and McDonald’s omega. For the sake of clarity and conceptual precision, please only use one term throughout the manuscript.

- The manuscript should be carefully proofread, as it contains several minor errors and awkward phrasings. A few examples:

o On p. 3, line 17, you wrote: “[…] encompasses both negative (e.g., depression, anxiety) and negative aspects […]”. This should be corrected to: “[…] encompasses both negative (e.g., depression, anxiety) and positive aspects […]”.

o You wrote: “Participants were free to accept or decline participation; no fee was given to any participant whatsoever.” Consider revising this to: “Participants did not receive any incentives for participation.”

o You wrote: “Findings supported that all 5 items were loaded into a single underlying factor in all six countries […]”. A clearer phrasing would be: “Results showed that all five items loaded onto a single latent factor in all six countries […]”.

References

Clarke, B., Alley, L. J., Ghai, S., Flake, J. K., Rohrer, J. M., Simmons, J. P., ... & Vazire, S. (2024). Looking our limitations in the eye: A call for more thorough and honest reporting of study limitations. Social and Personality Psychology Compass, 18(7), e12979. https://doi.org/10.1111/spc3.12979

Giner-Sorolla, R., Montoya, A. K., Reifman, A., Carpenter, T., Lewis Jr, N. A., Aberson, C. L., ... & Soderberg, C. (2024). Power to detect what? Considerations for planning and evaluating sample size. Personality and Social Psychology Review, 28(3), 276-301. https://doi.org/10.1177/10888683241228328

Goretzko, D., Siemund, K., & Sterner, P. (2023). Evaluating Model Fit of Measurement Models in Confirmatory Factor Analysis. Educational and Psychological Measurement. Advance online publication. https://doi.org/10.1177/00131644231163813

Gunn, H. J., Grimm, K. J., & Edwards, M. C. (2020). Evaluation of six effect size measures of measurement non-invariance for continuous outcomes. Structural Equation Modeling, 27(4), 503-514. https://doi.org/10.1080/10705511.2019.1689507

Groskurth, K., Bluemke, M., & Lechner, C. M. (2024). Why we need to abandon fixed cutoffs for goodness-of-fit indices: An extensive simulation and possible solutions. Behavior Research Methods, 56, 3891–3914. https://doi.org/10.3758/s13428-023-02193-3

Kline, R. B. (2023). Principles and practice of structural equation modeling (5th ed.). Guilford publications.

Kline, R. B. (2024). How to evaluate local fit (residuals) in large structural equation models. International Journal of Psychology, 59(6), 1293-1306. https://doi.org/10.1002/ijop.13252

Little, T. D., Slegers, D. W., & Card, N. A. (2006). A non-arbitrary method of identifying and scaling latent variables in SEM and MACS models. Structural Equation Modeling, 13(1), 59-72. https://doi.org/10.1207/s15328007sem1301_3

Marsh, H. W., Hau, K. T., & Wen, Z. (2004). In search of golden rules: Comment on hypothesis-testing approaches to setting cutoff values for fit indexes and dangers in overgeneralizing Hu and Bentler’s (1999) findings. Structural Equation Modeling, 11(3), 320-341. http://dx.doi.org/10.1207/s15328007sem1103_2

McNeish, D. (2023). Dynamic fit index cutoffs for categorical factor analysis with Likert-type, ordinal, or binary responses. American Psychologist, 78(9), 1061–1075. https://doi.org/10.1037/amp0001213

McNeish, D., & Wolf, M. G. (2023). Dynamic fit index cutoffs for confirmatory factor analysis models. Psychological Methods, 28(1), 61–88. https://doi.org/10.1037/met0000425

Nye, C. D. (2023). Reviewer resources: Confirmatory factor analysis. Organizational Research Methods, 26(4), 608-628. https://doi.org/10.1177/10944281221120541

Nye, C. D., Bradburn, J., Olenick, J., Bialko, C., & Drasgow, F. (2019). How big are my effects? Examining the magnitude of effect sizes in studies of measurement equivalence. Organizational Research Methods, 22(3), 678-709.

Sischka, P. E., Décieux, J. P., Mergener, A., Neufang, K. M., & Schmidt, A. F. (2022). The impact of forced answering and reactance on answering behavior in online surveys. Social Science Computer Review, 40(2), 405-425. https://doi.org/10.1177/0894439320907067

Sischka, P. E., Martin, G., Residori, C., Hammami, N., Page, N., Schnohr, C., & Cosma, A. (2025). Cross-national validation of the WHO–5 well-being index within adolescent populations: Findings from 43 countries. Assessment. Advance online publication. https://doi.org/10.1177/10731911241309452

Steiger, J. H. (2007). Understanding the limitations of global fit assessment in structural equation modeling. Personality and Individual Differences, 42(5), 893-898. https://doi.org/10.1016/j.paid.2006.09.017

Topp, C. W., Østergaard, S. D., Søndergaard, S., & Bech, P. (2015). The WHO-5 Well-Being Index: a systematic review of the literature. Psychotherapy and Psychosomatics, 84(3), 167-176. https://doi.org/10.1159/000376585

All other mentioned references here are listed in the manuscript.

---

## [Reviewer Report]

I appreciate the effort to conduct the study in six Arab countries. The scale is a widely used psychometric tool, yet it needs cultural validation.

Abstract

• Page 2, line 11-13: After acknowledging SWB as a culturally-dependent and context—driven construct, the 2nd line seems to be redundant. Also, the semicolon (;) should be replaced with a comma, and “It” should be lowercase. Consider removing the sentence. In line 16-17, the word subjective should be like this, “subjective” as the authors mentioning the construct- awkward phrasing of “construct subjective”; incorrect comma placement. The sentence is lengthy and hard to follow. Consider breaking it down into at least two sentences.

• Is this cross-country validation or cross-cultural validation?

• Do we assess cross-gender invariance using Pearson correlation?

• The last sentence of the Abstract should be revised because it conveys inaccurate information.

• Consider checking punctuations throughout the manuscript. A comma comes before the last name (Tunisia, Lebanon, Egypt, Jordan, Morocco, and Kuwait).

• Method: Among is used for relationships between multiple subjects/objects.

• We carried out (no hyphen) a cross-sectional, web-based study with/on a total of 3,247 young adults (aged 18-35 years, M=XX.XX, SD= XX.XX) from six Arab countries (i.e., Tunisia, Lebanon, Egypt, Jordan, Morocco, and Kuwait).

• Results: “mean scores” should be clarified as “Mean (M) and Standard Deviation (SD)” for accuracy.

• Confirmatory Factor Analyses“ should be singular (”Confirmatory Factor Analysis")

• “invariance was” ---- “invariance were” (plural subject).

• “strong significant” is redundant; use “strong and significant.”

• “Negative correlations have also been demonstrated”--- Should be in past tense for consistency (e.g., “were also found”).

• Check the reference style as suggested by the journal.

Introduction

Page 3:

Line 19: You have not established a case for SWB to be a complex construct other than saying it has both positive and negative aspects. Consider adding sentences explaining why SWB is complex.

Line 15: The concept of subjective well-being (SWB) encompasses both negative (e.g., depression, anxiety) and negative aspects (e.g., happiness, satisfaction, contentment. Shouldn’t, one be positive?

The well-established salutary impact of SWB on health several researchers called for including SWB as a measure of outcome of patient-centered mental healthcare. This sentence is awkward. Consider braking it into meaningful sentences.

As such, particular emphasis was placed in recent years on collecting self-rated SWB data in clinical settings, in the general population, as well as in research. Awkward phrasing and grammar. Has been placed… and SWB has been found to be closely…

…and several countries have already included SWB as a routine assessment to inform government decisions and public policy. It could be an independent sentence following the previous one.

Line 47-48: six-point scale?

Page 4: Line 8-9: We could found or we found?

Page 4: Line 14: Results of the same study or a different one? Provide citation.

Page 4: Line 22-29: This paragraph does not follow the validation studies conducted in Arabian contexts, rather suggests how robust the measure it. Consider moving this before discussing the Arabian versions.

Page 4: line 32: the subheading ‘Well-being in the Arab world’ sounds very dramatic and less academic at least in the current study. Alternatives could be, “Well-being in the Arab Contexts/cultures or “Well-being: Arab Perspectives’’ one or the other.

Page 4: line 35: This sentence requires a reference…a high burden of mental health problems (ref).

Page 4: mental disorder rates or mental disorder rates? Also, “Expected values” sounds unnatural in this context; “expected levels” is clearer.

“Arab Eastern Mediterranean countries” is an awkward phrasing; “Eastern Mediterranean Arab countries” is more natural.

Indeed, mental disorders rates exceeded the expected values in Arab Eastern Mediterranean countries, generating steadily increasing and higher than globally burden levels [51]. This sentence is very awkward. Consider revising it. Global levels? ‘Resulting in’ instead of generating? Expected to be on the rise or expected to rise?

“Such strategies are inappropriate and ineffective for dealing with the highly challenging conditions and deteriorating mental health that most of the Arab general populations are facing.” Are there better ways to frame this sentence? Also ‘Arab general populations’ or Arab population?

“Becomes urgently needed” or “is urgently needed”

Page 5, line 28-19: The vast majority of previous validation and adaptation studies of the WHO-5 were performed in Western countries with individualistic backgrounds. This claim requires a citation. Also, the use of ‘however’ in the following sentence does not mean anything. Use connecting sentence to smoothen the transition from the use of WHO-5 scale in Western countries to SWB being culturally dependent. Also what is with the punctuation ;? Then again the 3rd sentence (after ref 62 and 63) does not follow the 2nd sentence. The remaining paragraph provides evidence in favor of the scale being cross-culturally validated tool. How does the claim then make sense when the authors say Despite these data, the cross-cultural validity of well-being scales is still an unexplored question [65]? This entire paragraph needs to be rewritten focusing more on the gaps as to why the present study was conceptualized to being with.

North Africa: Tunisia and Morocco?

However, variations across genders may also be largely driven by methodological factors [71]. Justify this information with your aim.

Larger sample of participants? Is the 4th objective really an objective? Having a large same size is a strength of the study not an objective/aim.

All psychometric properties are important. So, the use of important psychometric properties seems awkward.

How many times do the authors need to state their objectives? See page 6 line 40-54

Can the authors remove the word indeed throughout the manuscript? I don’t see any reason to use that.

Are these references (75-77) indicating the use of snowball sampling and online recruitment in Arab countries?

All collaborators who collected data were asked to follow the ethical guidelines of their Institutional Review Board (IRB), acting either on the ethical approval received from their local IRBs or that of the Principal Investigators. What does this mean? Consider making it clear. Also, include individual IRB numbers.

The use of Instagram for collecting data requires a justification as this is a new trend. Did the authors use any hashtags? How did they engage prospective participants on Instagram?

Measures

• WHO-5 is a six-point Likert scale.

• Among elderly people or on elderly people?

• Columbia Suicidal Rating Scale: Is the omega (.79) only for adults not available for adolescents?

• Higher scores reflect more severe insomnia (ω = .82). What exactly does this mean?

• More information on the measures is required (e.g., more psychometric properties). Also, is the construct ‘suicide’ pancultural in Arab these countries? Just because they speak the same language does not mean the conceptualization of suicide, depression, and anxiety will be the same across all countries selected for the study. Justify the use of these measures.

CFA: Why was the threshold of VIF < 5 chosen and not VIF <10? Please justify.

Further analyses: The scale names have not been abbreviated earlier.

Results

3.2. Confirmatory Factor Analysis of the Arabic WHO-5

• Why did the df (from 5 to 4) change in the modified model? Usually, it changes when parameters are added.

• AVE values > 1 are mathematically impossible in FA as they indicate error in the calculation. AVE values range from 0 to 1.

• The model fit improves after adding a residual correlation. Therefore, the degree of improvement, especially RMSEA seems significant. Please explain the modification was justified.

• The perfect or near-perfect fit statistics (RMSEA CI for Morocco, CFI and TLI for Morocco, SRMR for Morocco) in the table (Table 1), seem positive, however, they should be examined carefully for potential overfitting or data issues.

• Tunisia (0.112) and Jordan (0.107) RMSEA values indicate poor fit, conflicting with other indices.

• Morocco: TLI (1.004), CFI (1.000), and RMSEA CI (<0.001–0.085) suggest overfit or data issues (small sample?).

• Path diagram shows unstandardized loadings (e.g., 1.00, 1.02, 1.05, F1 variance = 1.48), but the caption and table imply standardized loadings (≤ 1). Figure 1 caption (“Standardized Factor Loadings”) conflicts with the diagram’s unstandardized presentation.

• Error variances in the diagram (0.53, 0.43, etc.) don’t match expected values based on standardized loadings from the table (e.g., 0.2256 for 0.88: Tunisia. Error for 0.88 = 1 – 0.88² = 1 – 0.7744 = 0.2256. but the diiagram shows 0.53 for e1, which doesn’t match.

3.3. Measurement Invariance

• Were instead of was (see 1st sentence)

• P’s should always be italicized.

• Table 3: Why were Jordan and Morocco underlined?

• While the explanation of the mean score variance across countries seems beyond the scope of the study, however, it should be explained in the discussion to get a perspective with all potential explanations.

Discussion

• Page 13, line 38: This study is the first to explore the cross-country validity of the WHO-5… the WHO-5 Well-being Index?

• These high mean scores should sound warning bells for clinicians, researchers and policy-makers working in Arab settings; and further highlights that local culturally-sensitive strategies are urgently needed to address well-being issues among Arab young adults. ‘further highlight’. Also, how did the author make such an ‘urgent’ inference about culturally-sensitive strategies being needed for Arab young adults?

• Page 14, line 25: Analysis of the present study?

• Page 14, page 55: Please change WHO-5 Well-being scores thought out the manuscript. ‘WHO-5 Well-being scores’ is the established term.

• There is a repetition of measurement invariance in the discussion. Put all the relevant information together to discuss the invariance. Also, there is an overemphasis of gender invariance results. Keep it short.

• Subject-verb agreement: “invariance was supported” “invariance were supported” (plural subject).

• The authors did not report conducting measurement invariance across multiple groups (e.g., countries) and yet they claim it as a multigroup measurement invariance. They conducted a gender-wise invariance.

• It would be interesting to know why WHO-5 Well-being items show measurement invariance across gender in these countries. Include a few potential explanations supporting this notion.

• ‘In the same line’, write ‘similarly’

• The clause “and was extensively applied” should maintain parallel verb tense. Change to “and has been extensively applied.

---

## [Reviewer Report]

Great work done to validate a very easy and simple assessment tool that can be used across different settings and population. Well done! I would like to comment regarding the assessment tools used and wether the tools used were validated in Arabic in other populations before this study. For example the Columbia Suicide Rating Scale and Insomnia Severity Index, unclear whether the one validated in Lebanon was in Arabic or not. Where did the author get the translated version of the Arabic WHO-5 that was validated and used in the Lebanon population among elderly people. Were there any discrpancy in translation across the countries?

Nonetheless, great work and would be a huge research and clinical contribution the MIddle east region and globally.

---

## [Reviewer Report]

Background

The authors should consider re-structuring the introduction to make it more concise and clearly articulate the rationale for validating the WHO-5 Well-Being Index (WHO-5 SBW) in this specific study setting, despite previous validations of the Arabic version by two independent studies. It would be particularly helpful for readers unfamiliar with the nuances of the Arabic language to understand why a third validation is necessary.

To streamline the introduction, the authors may consider reducing or omitting the general definition of mental health, assuming that most readers will already be familiar with it.

Additionally, to help readers better understand the context of the study and the representativeness of the sample, the authors could include information on internet penetration rates and access to the specific social media platforms (Facebook, Instagram, WhatsApp) used for recruitment. Providing basic demographic information, such as the male-to-female ratio among 18–35-year-olds in the study sites and data on social media usage in these groups, would clarify whether the recruited sample can reasonably be considered representative of the general population.

Methods

1. Provide the ethical approval details of all the participating countries from which study participants were recruited.

2. Clarify how the authors ensured that no duplicate entries were made

3. The selected sample age group may be roughly categorised into late adolescence/emerging adulthood (18-24) and young adulthood (24-34 years). For concurrent validity analysis, were there results to indicate measurement invariance in all selected constructs between these two groups, because literature suggests variability in symptomatology and prevalence of constructs like anxiety between younger and older adults e.g in this study https://pubmed.ncbi.nlm.nih.gov/30538658/

Results

1. Provide study participants data as table 1. Supplementary material was not provided with this submission therefore it is not possible for reviewers to determine whether there was overrepresentation by one country and whether these overrepresentations may partially explain overall good fits in aggregated results but variabilities in country-level analyses. For example, the RMSEA indicated poor fit for Tunisia and Jordan, even though the CFA and TLI were excellent. Besides degrees of freedom, sample size may partially explain this finding, hence need to present these as main results.

2. The findings that the models improved after adding a correlation between residuals of items 1 and 4 suggest that these two items share additional variance beyond the main factor which was successfully modeled. Do the authors have any elaboration on this finding or on how future studies should handle these two items?

3. Is there any rationale for not providing country level analyses for concurrent validity?

Discussion

1. The conclusions that the findings of the study are of a general population cannot be made, unless clarifications are made in the introduction (or methods) of the structure of the general population from which these samples were drawn

2. Consider reviewing the argument in the second paragraph of the discussion for clarity. Are high SWB a positive or a negative finding? I suppose the authors meant to say “low”. Also in line with the authors argument in the introduction, it is counter-productive to compare general adult population findings with young adult population findings which were used in this study

---

## [Editor Report]

Can you please address the reviewers’ comments, particularly ensuring that you address the following:

• Provide a more nuanced description of the subjective and complexity of SWB as this will strengthen the need for the study, further

• Motivate the rationale for not performing cross-country measurement invariance, given the multinational nature of the study, including the inherent differences in cultural conceptualisation of well-being 

• Sample size justification 

• Clear outline of the evaluation plan, including reporting of all pre-requisite fit indices for viewers to make an independent and informed judgement 

• Whenever possible, kindly stratify the results per country 

• Kindly address the grammatical errors and typos throughout the manuscript 

• Ensure that the conclusions are in keeping with the study’s methodology 

• Kindly reference all important statements 

• Please provide details for ethical approvals across the countries

---

## [Reviewer Report]

Thank you for the opportunity to re-review this manuscript. While the manuscript has substantially improved, two key concerns remain unaddressed. Please find my detailed comments below.

As a brief note on style: I often provide numerous references in my reviews. These are intended to guide and inform the authors and should not necessarily be cited in the manuscript unless I explicitly recommend doing so.

Regarding sample size: I do not question that your sample size is sufficiently large to conduct the planned analyses. However, it would be helpful to briefly explain how the sample size of 3,247 participants was determined (e.g., why this number and not 2,000, 5,000, or 10,000?). In many observational studies, sample size is determined by practical constraints such as time, budget, or logistical feasibility. If this was the case in your study, you might add a sentence such as: “We collected data until we reached at least [number] observations, at which point the survey was closed,” or “The survey was open between [start date] and [end date], and all responses collected during that period were included in the analysis.” Moreover, since you are conducting not only a simple CFA but also measurement invariance testing across different groups, you could follow up with: “The resulting sample size was sufficiently large to provide adequate statistical power for all our analyses, including measurement invariance testing across groups (e.g., Meade et al., 2008).”

Regarding language and clarity: The manuscript still contains several grammatical errors and awkward phrasings. To enhance clarity and overall quality, I strongly recommend a thorough language revision. Ideally, this should involve a professional proofreader or a native English speaker with experience in academic writing. If that is not feasible, you might carefully use a language model such as ChatGPT (https://chatgpt.com/). For tips on academic use, see e.g., https://scitechedit.com/chatgpt-and-academic-writing/ (points 4 and 5). However, please use such tools cautiously—always reviewing the output to ensure it accurately reflects your intended meaning and maintains scientific rigor. If you choose to use ChatGPT, I suggest revising the manuscript section by section with a prompt like the following: “Please help me revise the following section of an academic manuscript for grammar, clarity, and style. It should remain formal and scientifically accurate. Avoid changing the meaning, but feel free to rephrase awkward or unclear sentences. Here is the text: [Insert section here].”

Lastly, there is still some room for methodological improvement (as noted in my previous review), such as using dynamic fit index thresholds for model evaluation, reporting effect sizes for measurement non-invariance, and examining group mean differences (e.g., by gender, age, or country) within a latent variable framework (see Dimitrov, 2006). While addressing these points would strengthen the manuscript, I believe they are not critical to the core conclusions and can be left as suggestions for future work.

Minor points

- Please revise the sentence “CFA indicated that fit of the one-factor model of the Arabic WHO-5 Well-being Index was acceptable” to: “Except for the RMSEA, most CFA model fit indices indicated that the fit of the one-factor model of the Arabic WHO-5 Well-being Index was acceptable.”

- Please change the heading “Measurement invariance and gender differences by gender and countries” to: “Measurement invariance and differences by gender, age, and country.”

- I recommend integrating the results of Table 4 directly into the text. If you choose to retain the table, please revise the heading to: “Manifest WHO-5 Well-being scores across countries.” Also, remove the sentence “Numbers in bold indicate significant p values,” as only one p-value is shown.

- In Table 5, remove the “1” on the diagonal and add a note to clarify the asterisks (e.g., Note. *** p < .001).

- Since you also conduct measurement invariance and mean differences analyses across age group, you should also provide reliability coefficients for these groups.

- -You wrote: “The composite reliability was 0.94 for the original (without correlation between residuals) and final (after adding the correlation between residuals of items 1 and 4) models.” Does this mean that you adjusted the McDonald’s omega calculation to reflect the change in model specification—specifically, by including the estimated residual covariance in the denominator of the formula?

References

Dimitrov, D. M. (2006). Comparing groups on latent variables: A structural equation modeling approach. Work, 26(4), 429-436. https://doi.org/10.3233/WOR-2006-00576

Meade, A. W., Johnson, E. C., & Braddy, P. W. (2008). Power and sensitivity of alternative fit indices in tests of measurement invariance. Journal of Applied Psychology, 93(3), 568–592. https://doi.org/10.1037/0021-9010.93.3.568

---

## [Reviewer Report]

The authors have adequately addressed the reviewers' comments that directly impact the study’s interpretability and have provided the requisite supplementary materials.

---

## [Reviewer Report]

Thank you for the opportunity to re-review this manuscript. The authors have adequately addressed all of my previous comments. I have no further concerns regarding the manuscript in its current form.